# A Study Exploring the Implementation of an Equine Assisted Intervention for Young People with Mental Health and Behavioural Issues

Ann Hemingway 

Faculty of Health and Social Sciences, Bournemouth University, Poole BH12 5BB, UK;
ahemingway@bournemouth.ac.uk; Tel.: +44-012-0296-2796

**Abstract:** This paper presents the findings from a study of an equine assisted intervention (EAI), which is currently referred over 150 predominantly young people with mental health and behavioural problems each year. The young people are referred to this intervention when other services such as Child & Adolescent Mental Health Services (CAMHS) are not effective. Performing an exploratory study of implementation may be indicated when, there are few previously published studies or existing data using a specific intervention technique. This study showed some positive changes for participants across eight dimensions including; assertiveness, engagement with learning, calmness, planning, taking responsibility, empathy, communication and focus and perseverance. The equine intervention literature has shown mixed results across a variety of study designs and target groups, in terms of the gold standard of evidence, randomised controlled studies however the evidence currently is very limited. This study used a non-randomised sample, no control group and an unstandardised measurement filled out by those who refer young people to the intervention (social workers and teachers). The outcomes however from this exploratory study would suggest that a randomised control trial may be warranted and achievable.

**Keywords:** equine assisted; young people; mental health; behaviour

## 1. Introduction

This paper presents the findings from an exploratory study of an equine assisted intervention (EAI) which is referred over 150 predominantly young people with mental health and behavioural problems each year for whom other options such as referral to Child & Adolescent Mental Health Services have not been effective. There is some limited evidence that behavioural issues in young people may be impacted upon by equine assisted interventions. Performing an exploratory study of implementation may be indicated when, there are few previously published studies or existing data using a specific intervention technique [1].

## 2. Background to Equine Assisted Interventions (EAIs)

The equine intervention literature has shown mixed results across a variety of study designs and target groups, in terms of the gold standard of evidence, randomised controlled studies, however the evidence currently is limited.

There have been studies showing positive impacts through EAIs with those with disabilities, or illnesses either physical or mental [2–12], or individuals with eating disorders [13]. In addition, the potential benefits of equine assisted psychotherapy or experiential therapy have also been studied although the outcomes have been mixed in terms of the effectiveness of the interventions with some studies showing positive results [14–18] and some no effect [19,20].

Exploratory qualitative research has identified particular behaviours and emotions that may be experienced from interaction with horses and these include trust, motivation, patience respect and empathy and reductions in violent behaviour [21,22].

Pendry and Roeter [23] published a randomised controlled trial focused on evaluating the effectiveness of an EAI using natural horsemanship techniques on improving child social competence. Their findings reported moderately significant improvements in the social competence of 5th to 8thgrade children following the intervention. In 2014 Hauge et al. [24], reported on a study using a waiting list cross over design and a four-month intervention with horses with adolescents aged 12–15. The intervention group reported a significant increase in perceived social support compared with the control group. In 2015 Boshoff et al. [25], published an experimental design study, which measured subjective well-being, problem focused coping, and emotion focused coping and showed some positive change in young men in a custodial school.

A systematic review of equine assisted interventions [26] on psychological outcomes found that the quasi-experimental design studies included in their review (n = 7) showed improvements in social and behavioural outcomes thereby suggesting a potential role for equine assisted interventions in developing socially appropriate responses.

Nimer & Lundahl [27] published a meta-analysis of animal assisted therapy for young people which showed a moderate positive affect size in young people with autism-spectrum symptoms, medical difficulties, behavioural problems, and emotional well-being issues. While recent systematic reviews highlighted the need for studies in this area to offer detailed insights into the description of the intervention offered and use reliable measures as well as detailed qualitative exploration [28,29].

Most recently three systematic reviews have been published [30,31] focused on the impact of equine assisted therapy on physical symptoms in adults living with disabilities and the impact on those living with autism and schizophrenia respectively. All these reviews found that studies lacked rigour, randomisation and adequate sample size however those studies that have been published do report positive outcomes across a range of physical, behavioural and social areas. The autism review also commented on the variety of interventions offered and the lack of clarity in relation to what was included in the intervention with large varieties in experience for participants noted.

In addition one pilot study has been undertaken looking at the effects of EAIs and therapy on resting brain state function in attention-deficit/hyperactivity disorder [32]. This study found that the intervention was associated with short range functional connectivity in the regions of the brain related to the behavioural inhibition system, which are associated with symptom improvement in this group.

Overall research to date therefore has found that the interaction between humans and horses is an emotional experience, which may have an impact on behavior in young people [26,27], however, how this mechanism may work and further exploratory studies to underpin the potential for future randomized controlled trials to test the effectiveness of EAIs are lacking from the literature.

## 3. The EAI under study

This intervention is offered by a charity which operates in the South of England and is referred over 150 people every year by Schools & Pupil Referral Units, Children's Services, National Health Service (NHS, UK) Mental Health Services, Troubled Families, Offender Services and other specialist agencies such as charities working with Domestic Violence or Drug and Alcohol Services. The young people referred have typically 2–4 issues from the list below and they are referred because they are 'stuck' or disengaged from talk-based support. Some young people who are referred from Child & Adolescent Mental Health Services may have a formal diagnosis of a mental health problem (35% of referrals), however the majority are referred by social workers or teachers who report primarily the reason for referral in behavioural or social terms. All of the diagnosis/terms used are included in this list.

- Attention Deficit Hyper-activity Disorder
- Autism Spectrum Disorder

- Anxiety/depression/self-harm
- Bullying, aggression, anger management issues
- Domestic violence/neglect/abuse
- Bereavement
- Conduct disorder

The intervention uses the principles of the Parelli Natural Horsemanship program as its philosophical underpinning and structure [33]. This approach is based on learning calmness, cooperation and partnership development through natural horsemanship skills. At this introductory level this involves 'playing' with specially trained horses inviting them to respond to requests with the young person on the ground and the horse on a loose rope or at liberty.

The horses are highly trained but will only engage with the young person if they can gain the horses trust as a sentient being and partner in this activity through appropriate actions and behaviour [34]. These games help to establish a simple yet comprehensive communication between horses and humans. However in order to be a partner the human needs to use clear, phased assertive communication and control their body language and energy in an assertive, non-aggressive way. Rather than using the horse as a passive adjunct or emotional 'mirror' to a therapeutic intervention as in many other EAIs these horses teach the human participants through their responses while playing these games.

The learning is facilitated by a specialist facilitator and the students are taught how to play the seven 'games' [33] with the horse.

The course takes place in an indoor arena over 10 hours in five two-hour sessions over the course of a week and costs £750 per participant.

The games taught are:

1) *The Friendly game* (creating relaxation through touch, grooming, hanging out).
2) *The Porcupine game* (moving the horse's feet through using steady pressure, touching the horse).
3) *The Driving game* (moving the horse's feet through rhythmic pressure, not touching the horse).
4) *The Yo-yo game* (moving the horse backwards and forwards).
5) *The Circling game* (asking the horse to travel around you on the circle).
6) *The Sideways game* (asking the horse to move sideways).
7) *The Squeeze game* (asking the horse to go through, under or over something [33]).

## 4. Materials and Methods

The methods used to inform this exploratory study included:

- Analysis of referral data from the charity, (n=155) see Table 1.
- Analysis of before and after scores on the 'star chart' Figure 1 for participants completed by referrers (n = 155) see Tables 2 and 3.
- Analysis of after only scores for participants completed by referrers see Table 4 (n = 155).

The different elements of the study will be presented here. Both the positive and negative aspects of the course in relation to how feasible it is to further research this intervention using a randomised controlled trial (RCT) and to offer the course to a wider population will be considered.

*Ethical Considerations*

Ethical permission was gained from the researchers employing universities ethics panel (BU REF 8750). The star chart tool and further assessment tool was completed electronically by the referrer to the intervention. The scores were then stored on an Excel spread sheet by charity staff with each participant being allocated a code number so the researcher could not identify the young person or the referrer, This registered charity undertakes risk assessments for all participants who are never left unsupervised with the horses. The horses are all observed for possible stress/distress continuously throughout the course activities informed by the ethogram of horse behaviour [34] activities would

cease immediately if any observations of this were made. All the horses who engage in these courses with young people are kept outside in a natural environment (hedges, trees other horses) in friendship groups with access to shelter if they want it.

## 5. Results

### 5.1. Methods

The mean age of the subjects was 12.55 years (st-d 3.269) with 97 girls and 58 boys included in the sample. The sample constitutes a convenience sample of 8–18 year olds accessing the course over a two-year period to May 2018 for whom full data and consent were obtained. In this study before and after measures were completed by the young person's referrer to the course n = 155 completed scales were included in the analysis (see Figure 1.). The star chart (Figure 1.) has been developed by the charity running this intervention. The referrer (normally a social worker or school teacher) rates the young person from zero stuck, to four independently using the skills being assessed. These are calmness, assertiveness, planning, empathy, communication, focus and perseverance, engagement and confidence as a learner and taking responsibility for their actions. These areas arose as potential outcomes from early qualitative studies of this intervention [22].

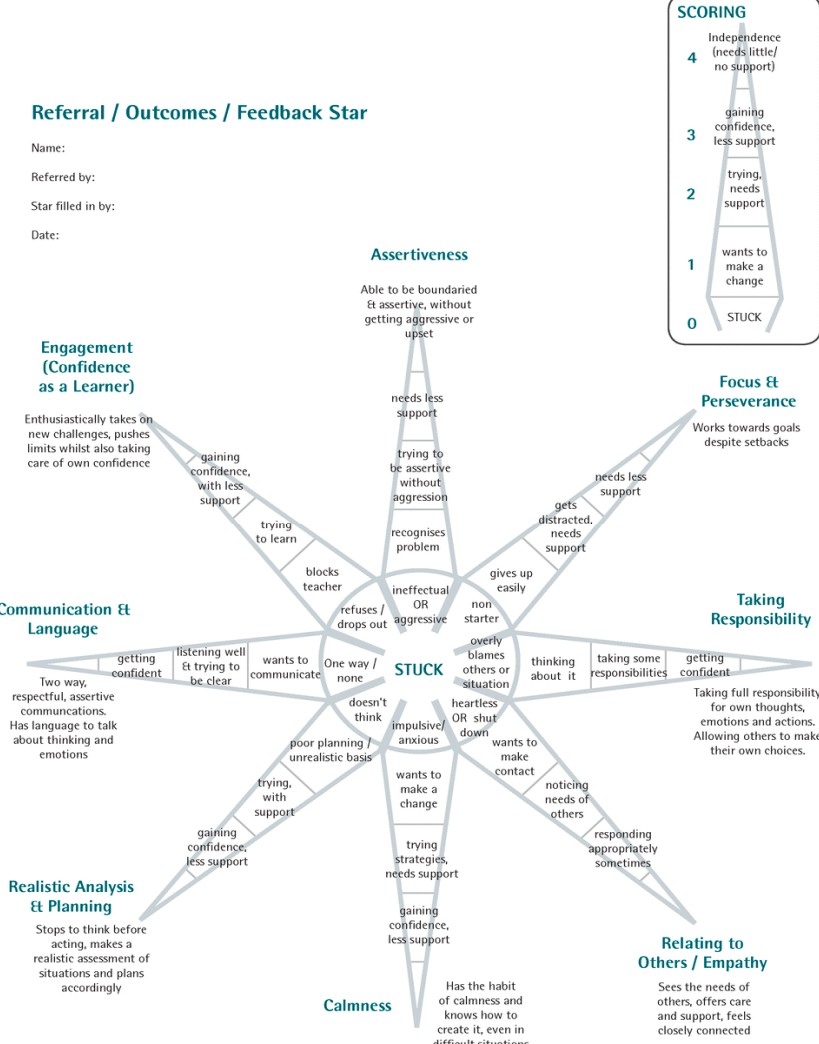

**Figure 1.** The Star Chart.

The reasons for referral to this intervention are shown below in Table 1.

**Table 1.** Reason for Referral.

| Main Reason for Referral | Anxiety/ Depression | ADHD | Bullying/Angry/ Aggressive/Violent | Lacks Confidence/ Bullied/Lonely | Witness to Domestic Violence/Abuse | Not Attending School/ Excluded From School | Self-Harm/ Suicidal | Relationship Issues/ Attachment Disorder |
|---|---|---|---|---|---|---|---|---|
| n = 155 | 18 | 15 | 53 | 27 | 11 | 10 | 13 | 8 |

Pre-test scores were analysed using Cronbach's Alpha to assess the tools internal reliability the score was 0.9 indicating high internal reliability.

Data was analysed using SPSS and a non-parametric related sample sign test which showed statistically significant improvements between the pre and post-test scores for eight dimensions with a significance level of $p < 0.001$ for all dimensions. Included here in Table 2 are the numbers of positive, negative and no change scores for the sample (total number differences are due to missing scores for that dimension for that young person).

**Table 2.** Positive Negative and No Change Scores for all Participants ($p < 0.001$).

| Star Chart Skills | Number of Participants with a Positive Score Change | Number of Participants with a Negative Score Change | Number of Participants with No Score Change | Total Number of Scores in Analysis for Each Skill |
|---|---|---|---|---|
| Realistic Planning | 105 | 15 | 33 | 153 |
| Assertiveness | 107 | 12 | 35 | 154 |
| Communication | 102 | 25 | 27 | 154 |
| Calmness | 111 | 15 | 28 | 154 |
| Engagement (as a learner) | 103 | 13 | 38 | 154 |
| Focus and Perseverance | 100 | 9 | 45 | 154 |
| Empathy | 99 | 19 | 36 | 154 |

Overall scores for all eight dimensions of the star chart improved for 141 of the participants with 14 experiencing a worsening of their overall score two months after completing the course. On examination of each of these individuals there was no consistent area which worsened across this group. All of the 14 did have some dimensions where they had improved their score or remained at the same score pre and post see Table 3.

The average increase in score for the 141 participants with an improved score was 1.17 on the scale of zero stuck to four independently using the skill being assessed.

**Table 3.** Characteristics of those with negative overall score after the intervention.

| Age | Gender | Reason for Referral |
|---|---|---|
| 11 | F | ADHD |
| 13 | F | Bullied/alcoholic step father |
| 8 | F | Poor parenting |
| 8 | M | Violent |
| 15 | F | Lacks confidence |
| 16 | F | Self-harm |
| 9 | F | Anxiety |
| Not known | F | Poor school attendance/Bullying |
| 11 | M | Agressive |
| 13 | F | Relationship Issues |
| 13 | F | Bullied/Isolated |
| 13 | F | Aggressive |
| 10 | F | Physically abusive at home |
| 9 | M | ADHD |

Referrers also scored participants two months after completion of the course on each of the following areas. The scale ran from −1 worse, 0 no change, 1 positive change. The results are presented here in Table 4.

**Table 4.** Follow Up Scores.

| Further Outcomes Assessed by Referrer | Engagement with Education | Problem Behaviours | Relationships | Sense of Identity |
|---|---|---|---|---|
| Worse | 8 | 7 | 5 | 5 |
| No Change | 25 | 36 | 31 | 38 |
| Positive Change | 122 | 112 | 119 | 112 |
| Number in analysis | 155 | 155 | 155 | 155 |

In addition, the following areas were considered specifically to offer an exploration of whether a randomised controlled trial could be undertaken on this intervention [1]. The answers to these questions were found from the analysis of the referral data and completion data for the charity and through interviews with the CEO.

*5.2. Implementation*

Over the period of data collection for this study 2016–2018 155 young people aged 8–18 have completed the intervention with an evaluation tool (before they start the program and then two months after they finished the program) completed by their referrer. An additional five young people did not complete the intervention during this time; one due to physical illness; the other four due to underlying family issues. It would appear from attendance records that once younger people engage with this intervention they do finish it, other than the five outlined above there were no other non-completions during this period.

Referrals are increasing with referrers from multiple agencies, as outlined sending those people for whom talk based interventions are not working. The charity itself raises the funds to offer this opportunity to young people and more recently to older adults referred predominantly by local mental health providers. In the long term this would not be sustainable for a larger group of participants and local commissioning of this provision would need to be negotiated through the health services (UK NHS) commissioning group.

The program which we are focusing on in this paper (Restart) forms the majority of the activity undertaken by this charity. There are also plans to do lighter touch work with a wider group in the future in response to demand from a variety of referrers who feel that young people may benefit from a shorter course earlier in their difficulties, or on-going support to maintain behaviour change after they complete the course.

Once again there may be an issue of supply and demand and future secure funding is also an issue in relation to further development of this provision. The main resource issues are the highly trained instructors in natural horsemanship (individuals with typically 10+ years of training). In addition these individuals need further development to facilitate this intervention from the charity. A physical site for the intervention is needed, and having a yard and an indoor school with a special soft surface is required in order for the charity to function all year round. All of which are both expensive to fund and unusual to find around more densely populated areas. This may cause problems for attendance for those without transport.

According to the charity CEO this intervention has not substantially changed since the first month, however the relationships within the family of those who are referred and the resilience skills within the family as a whole have been identified as very important by the young people, course facilitators and the referrers. Therefore the charity is now undertaking work with troubled families, foster families and the teams around the family to try to positively impact on these processes. Referrals for this provision are coming from the Local Authority Troubled Families team.

Individuals tend to get referred to this intervention when they are stuck in treatment being 'bounced' between services repeatedly with no progress being made. The young people referred are often in contact with multiple agencies and are referred with multiple issues as explained earlier.

This charity has now replicated its programs across four counties. Additional facilitators have been trained by the charity to aid replication, recruited primarily from those who already hold instructor status with Parelli Natural Horsemanship.

This has created challenges however in terms of replicating the exact provision and the evaluation of the service as busy social workers and teachers find repeated measurement tools onerous. In addition, there is a limited pool of horses and instructors as mentioned previously.

## 6. Discussion

The need locally to work with participants referred to this intervention with a relatively wide variety of issues is harder to accommodate within a traditional RCT design as normally in a trial the reasons for referral are fairly narrow in order to facilitate comparison with a clear alternative treatment, 'treatment as usual', or no treatment comparison group. However, this could be done if clearly justified in the study design. In addition, if one focused on a particular group out of the list of reasons for referral it is possible that the numbers of participants recruited successfully to a trial would be small. If data were collected over a two to three year period to enable a power calculation to be statistically robust enough for effective analysis and reliable outcome measurement an RCT focusing on the most common group referred is still feasible at around 150 participants accounting for a ten percent drop out or non-completion of measurement tools.

A key element of this intervention would appear to be that the facilitator and the student are guided by the same principles used for teaching horses, which focus on reading body language and responding appropriately. The course teaches through simulation rather than explanation and uses non-verbal methods (through rehearsal) to impact on the thoughts and emotions of the students. Crucially the students rehearse embodied communication from the start.

Embodied pedagogy is defined as an educational program which joins body and mind in a physical and mental act of knowledge and skill construction [35]. Embodiment transcends linguistic differences and would appear to be a universal language with other mammals possibly available to us as human mammals for learning as a pre-language ability [36]. As the participants rehearse the embodied skills required to enable them to 'play' with the horse asking them to do things and becoming successful the 'learning' is rehearsed and reinforced through their bodies enabling them to feel calm and assertive. Arguably this embodied or pre-speech capability to communicate with others as with young children and as in this case communicate and learn with a non-human animal using 'internatural' embodied mechanisms may offer an opportunity to interrupt previous potentially maladaptive habits developed to deal with communication and potential conflict [37]. This learning process may offer a safe opportunity to rehearse communication.

It is particularly important when evaluating an intervention to consider the qualities of the intervention that render it effective. Students on this course have to try to understand the point of view of another being, the horse as a prey animal, in order to be effective when communicating with them through their bodies, these are qualities which Prior & Mason [38] highlighted when considering the evidence so far on what works to engage young people. As a result of this course participants may learn to 'listen' to another 'being' through their body language and rehearse this skill throughout the course.

It is interesting that an RCT undertaken with a mindfulness program [39] focused on adolescents showed some significant positive improvements in self-esteem, mindfulness, resilience and psychological inflexibility for young people with mixed mental health problems. There seems to be a consensus that mindfulness is a learned skill that enhances self-management [40] the qualities of which may be further embedded by learning natural horsemanship skills which require a state of active, assertive, open attention on the present moment. This could perhaps be described as mindfulness with

feedback from the horse which informs participants when they are being successful, as in when they have successfully communicated with the horse. This RCT was undertaken focused on a mindfulness group intervention which was talk based however, which may have been unacceptable to the group of young people with which the equine assisted intervention under study here is engaging.

The course under study here costs £750 per participant for ten hours spread over one week. In 2011 a report from the UK Department of Health [41] outlined the potential cost savings of social and emotional learning programmes to prevent or help reduce conduct problems in childhood. This report has since been followed up by a report from Public Health England in 2012 [42]. The prevention activities represented in these reports provide a ROI (return on investment) that vary between £1.26 and £39.11 per £1 spent on these activities. Where it is possible to estimate impacts on quality of life, all of the interventions appear to be cost-effective, with a cost per quality adjusted life year (QALY) gained below £20,000. This is the same threshold as used by NICE (National Institute for Health Excellence, UK). They estimated that programmes which help children and young people to recognise and manage emotions, plan goals, appreciate the perspectives of others, maintain positive relationships, and handle interpersonal situations constructively significantly improve social and emotional skills, attitudes, behaviour and academic performance [42]. In the reports they outlined the potential 'cumulative pay offs per child' across a wide range of services such as education, NHS, Social Services, Criminal Justice and the Voluntary Sector and they estimated that such interventions by year ten post intervention accumulate savings across services of £10,032 (acknowledging that this economic calculation is somewhat dated). These savings are gained primarily through the potential costs of crime, mental illness and lower lifetime earnings when conduct problems for children and young people are left un-treated.

The costs for CAMHS referral in 2016 for community care for one year were £2518 [43]. It is worth considering however that CAMHS are referring more than one third of the young people engaging with this charity who are not responding to or engaging with CAMHS talk-based interventions. Therefore this cost comparison also has limited relevance for this particular group.

The people who normally attend this course have a variety of different behavioural and mental health issues as outlined earlier many of which are associated with abuse and neglect or witnessing violence in the home. Interestingly through evaluation of the course they all leave with similar benefits the most consistent and strong of which is calmness in our evaluations so far thus enabling participants to re-engage with education and relationships. With those who refer to the course (primarily social workers) reporting rapid and effective changes in behaviour as one participant said following completion of the course 'I feel reborn'. Does using embodied inter-species interventions offer the opportunity for those for whom 'talk' based interventions are not working to gain therapeutic benefit?

Considering the insights presented here it is clear that there are potential issues in rolling out this intervention on a large population-based scale. Primarily relating to the resources required both in terms of horse and human instructors but also the physical spaces needed as already outlined and the progress in commissioning and funding the intervention required for longer term sustainability.

## 7. Limitations

This study did not randomise to the intervention or to a control group and used a convenience sample collected over a two year period. As such any possible impacts need further testing using a randomised sample. It is not clear from this study whether any potential outcomes relate to the overall intervention itself with the horses or the relationship developed with the course facilitator or the horse during the process. Indeed changes could relate to the passage of time between the intervention occurring and the follow up measure being undertaken (two months) or in relation to the age of the participants; it is possible that changes occurred due to development rather than the intervention itself.

In addition the tool used for the before and after scores has not been validated for use with this sample and has been developed by this charity for use with their participants; the dimensions of the tool (Figure 1.) emerged from previous qualitative research undertaken with participants of this

intervention [26]. However, this study suggests that it may be worthwhile to validate this instrument against other well being measures used with this group.

No statistically significant findings were discovered to help explain the 'no change' scores for some participants. A randomised sample and further statistical analysis is required to help further explain this.

## 8. Conclusions

Starting to think about how to help our young people develop or learn the path to wellbeing through learning embodied skills is a new way of thinking about this area. This intervention would lend itself to a more robust study on effectiveness using a randomised controlled trial particularly in relation to ensuring statistical reliability. Access to school attendance and achievement records could also be explored pre and post intervention. Further study of this intervention would also benefit from detailed economic analysis of the potential future costs of not treating young people for whom violent behaviour and lack of success in talk-based treatment are primary problems.

**Author Contributions:** The author is responsible for study design, data collection, data analysis and preparation of this paper.

**Funding:** This research received no external funding.

**Conflicts of Interest:** The author declares no conflict of interest.

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
