# Peer review of "A Study Exploring the Implementation of an Equine Assisted Intervention for Young People with Mental Health and Behavioural Issues"

_2571-8800, doi:10.3390/j2020017_

Reviewer 1 Report

This manuscript needs a substantial re-write, as in its current format it is very difficult to read. In current format, I cannot recommend it for publication. I would also suggest it be submitted to a journal specialising in youth mental health or psychotherapeutic interventions.

Title should reflect what the intervention is being tested for and the study design, i.e. for what group of people, for what outcome, and the design of the study.

Introduction is quite jagged. Unusual that it starts off by stating what this paper is about. More information about what a feasibility study actually involves is needed here.

Eight areas of feasibility focus: Acceptability, demand, implementation, practicality, adaptation, integration, expansion and limited efficacy testing. I would question whether this is actually a feasibility study at all – the manuscript doesn’t match the NIHR guidelines/parameters for feasibility studies of interventions - https://www.nihr.ac.uk/funding-and-support/documents/funding-for-research-studies/research-programmes/PGfAR/CCF-PGfAR-Feasibility-and-Pilot-studies.pdf.

Good to see that systematic reviews cited but no critical evaluation of similar research looking at EAI. Limited discussion about WHY exactly EAI may help.

Do the children who participated in the intervention have a suspected or confirmed diagnosis of the issues listed (lines 91-96)?

Good description of the intervention, but no information about the period of delivery, e.g. was it over a week, a month etc.

Materials and methods: this section is far too short to provide any information about the feasibility study.

·         “Analysis of before and after scores for participants completed by referrers” – scores on what measure? What was the outcome measure?

·         No description of any outcome measures or how the eight areas of feasibility focus were measured

·         No information about the design

·         No information about ethical approval and considerations

·         No information about data analysis

·         No information about participant referral into study, e.g. the procedures

Results:

·         “155 young people aged 8-18 have completed the intervention with an evaluation tool” – what are the demographics of the sample? E.g. age, gender, their presenting health issues. What is the evaluation tool? More information needed.

·         “An additional 5 young people did not complete the intervention during this time.” – what were the reasons for this?

·         “Referrals are currently increasing with referrers from multiple agencies as outlined sending those people for whom talk based interventions are not working” – this makes it sound like the study is currently ongoing, it’s not finished yet?

·         All the information provided under the “acceptability” heading doesn’t actually assess how acceptable the intervention is to the young people receiving it.

·         Demand – “demand is outstripping supply” – how does the reader know this? What’s the waiting list like? What’s the maximum number of young people that can be on the intervention at any one time?

·         Implementation – none of the information here actually looks at the implementation of the intervention. Please look at implementation science papers. E.g. what is helping facilitate the intervention, what are barriers to implementing it?

·         Practicality – practicality in terms of what? It sounds like it’s in terms of location/ What about from a young person’s perspective?  

·         Adaptation – not sure what this exactly means. Doesn’t it overlap with implementation?

·         Integration- integration with what, exactly? Not sure what this means.

·         Expansion – again, does this link with implementation and how it can be expanded into other equine centres?

·         Limited-efficacy testing

·         “in this study before and after measures were completed by the young person`s referrer to the course n=155 completed scales were included in the analysis” – again, in the text it’s not described what was measured. I can see from Figure 1 what was measured – is this a valid outcome measure? Developed for this evaluation – how was it developed? how do we know it’s got good validity and reliability? More information needed about why a Star Chart was used.

·         Ethical considerations is discussed in Results – this should be much earlier in the manuscript. Up to this point, I was assuming the study wasn’t ethically reviewed at all. L9ikewise this is the first mention of participant demographics – this should have been first thing in the results section.

·         Table 1 – could participants be referred for more than one of the reasons?

·         Not sure why Table 2 is included – could this statistic not have been in text only?

·         It’s only apparent here that a pre-post design was performed. At what time points was the outcome measure administered? The design should be mentioned earlier on in the manuscript.

·         Tables 3 and 4 should show the actual data from the outcome measure, not the numbers having a positive change etc. Test statistics should also be reported.

·         “with 14 experiencing a worsening of their overall score two months after completing the course” – were harms/adverse events arising from the intervention measured?

·         Table 4 shows Characteristics of those with negative overall score after the intervention – how did this align with those for whom had a positive overall score? Was there a particular type of person the EAI didn’t work for?

Discussion

·         “The need locally to work with participants referred to this intervention with a relatively wide variety of issues is hard to accommodate within a traditional RCT design as normally in a trial the reasons for referral are narrow in order to facilitate comparison with a clear alternative treatment, `treatment as usual`, or no treatment comparison group.” – this should be mentioned far earlier in the manuscript as I have been reading this manuscript and wondering why an RCT or even a better-designed pre-post design was not used? It’s fine for an RCT to have participants with many health issues participating, providing it’s clear and has a strong rationale for doing so.

·         Unsure how this could inform a full trial because the outcome measures reported here do not allow you to undertake power calculations, or understand the feasibility of the trial design and procedures etc.

·         “This intervention does not use cognitive (talking or classroom based) approaches as is generally the default within interventions for this group currently” – but there is an element of cognitive based approaches in EAI? It involves some interaction and communication between children, horses and facilitators?

·         How do the skills learnt in the EAI translate into non-EAI settings? Was any qualitative research done to undertake process evaluation of the intervention?

·         “The course under study here costs £750 per participant for ten hours spread over one week” – it should be mentioned earlier on in the manuscript how much the intervention cost, and the impact of this intervention upon other outcomes e.g. health service use, SENCO time etc

·         Limitations need further consideration

Author Response

Please find attached a detailed response to all of your useful comments please note I have used track changes in the paper for all revisions and have made comments in the text to highlight where the specific changes are in relation to each reviewers comments. Many thanks yours ann hemingway

Reviewer 2 Report

The authors of this paper have made an admirable effort to gain further information on the effects of equine assisted therapy. However there are  number of weaknesses that invalidate this as a scientific research project. Most importantly, they have used an unstandardised instrument to assess participant changes. As this instrument has not been tested or validated, it can at best be seen as supplementary evidence of changes. Also the authors have failed in either their design or their reporting, or both, to acknowledge other factors that could be contributing to participant changes.For example, It could be the therapuetic relationship between therapist/equine assisted coach and participant that has caused the change. Because of the lack of a control group this is particularly important. Also because the protocol of working with the horses is not described, it is unclear exactly what exactly is being measured.

Nevertheless, this paper has worthwhile contributions to the literature. I recommend the paper should be rewritten with more attention to the limitations and will less positive bias towards the results. Use of more scientific language throughout is also required to meet the standard of publication.

Author Response

Many thanks for your useful comments I have uploaded a detailed response and have used track changes to highlight all changes in the paper. I have also used comments in the text to show where specific responses are to your comments. Many thanks yours ann hemingway

Round  2

Reviewer 1 Report

-  Some typos and grammatical errors throughout - please review manuscript and make changes. For example, some text is in italics but unsure why. Another example is [3??] in the text.

- The response to reviewers is not what I would typically see as a reviewer - it has made it more difficult to re-review the manuscript and identify the changes made. Having read it, I'm unsure all my previous comments have been satisfactorily addressed, e.g. still nothing in introduction about why EAI might be helpful for young people, or information about how the intervention was delivered.

- Please can the response to reviewers be re-submitted which clearly states how each reviewer's comment was met, rather than signposting towards page numbers. I have found it very difficult to figure out how the reviewer comments were responded to.

- The title of the paper still doesn't reflect what the intervention is actually for, and for whom.

- Still not sure why Table 2 is there given that all the text within Table 2 is presented in the text?

- Again I would argue that 'acceptability' wasn't explored. Sounds more like compliance or adherence to intervention. 

- Line 544 ‘currently’ – what is ‘currently’, what point in time is this?

- Still no information about ethical approval. I gather this was not needed for this intervention - but it still needs to be explained why so.

Author Response

Many thanks I have not included comments in the text in relation to your reviewer comments and have included the line numbers here. I hope that is clearer in terms of the changes I have made.

Reviewer 2 Report

I can see that it has been changed from 'exploratory' to 'feasibility'. Because I cant see comments in my version, it is difficult to see what changes have been made.

There is still too much positive bias. For example in the Limitations you write

As such the positive impact needs further 323 testing using a randomised sample.

You cannot say there is a positive impact - positive impact of what? The design does not really allow this conclusion. You also need to mention that improvement in scores may be co-indicental with the treatment or may be due to maturation effects etc. This would show that you know there is more work to do to make your findings conclusive.

Also in regard to the instrument - you write

In addition the tool used for the before and after scores has not  been validated for use with this sample.

Has it been validated in any population?

The best you can say is that this study demonstrates that it results show it is worthwhile validating this instrument against other well being measures.

Limitations should be much lengthier in this paper. I note you haven't added my suggestion that therapuetic effects could be from other causes than working with the horses.

I think you also need to add limitations into your abstract (non-randomised, no control group and using an unstandardised measurement filled out by carers). and be specific about what is statistically significant.

The more I think about this paper, the more I think that your findings could be about the potential value of your STAR measure rather than the effectiveness of the treatment. Your building blocks are missing but your assessment shows that this measure may have merit. I realise that would be a significant rewrite though so am not asking for this.

Author Response

Many thanks please see attached file for detailed responses to the reviewers comments.

Round  3

Reviewer 1 Report

Thank you for clarifying how previous comments were met. However there are still a number of places in the manuscript that would benefit from proofreading. For example, sentence “Pre-test scores were analysed using cronbach`s alpha in order to assess the tools internal reliability the score was 0.9 indicating the tool has high internal reliability” does not read well, and Cronbach’s alpha should be capitalised. Likewise sometimes ‘table’ is capitalised and sometimes it isn’t – regardless of where it is in sentence. “The scores were then stored on an excel spread sheet by charity staff” – Excel should be capitalised. Some text under limitations is highlighted grey. These are just some examples – I haven’t highlighted all errors here. Please take time to thoroughly proof-read the document.

“The course takes place in an indoor arena over 10 hours in five two hour sessions over the course of a week and costs £750 per participant” is in red colour font, any reason for this?

“Data was analysed using SPSS and a non-parametric related sample sign test which showed statistically significant improvements between the pre and post-test scores for all eight dimensions with a significance level of p<0.001.” – all test statistics on eight dimensions should be reported in full in the paper, not just the significance level.

Is the referencing correct? The first references are [26, 27] – shouldn’t it be starting at [1] if you’re using Vancouver referencing? Please check journal instructions for authors.

Reviewer 2 Report

Changes have been addressed.

Changes needed to the sentence about maturation - where the definition of maturation in this context appears to have been mis-understood. Maturation refers to any variable that may change over the time of the experiment.